# Antioxidant Efficacy of Rosemary Extract in Improving the Oxidative Stability of Rapeseed Oil during Storage

**DOI:** 10.3390/foods12193583

**Published:** 2023-09-27

**Authors:** Mimi Guo, Liping Yang, Xiujuan Li, Huan Tang, Xin Li, Yalin Xue, Zhangqun Duan

**Affiliations:** Institute of Cereal & Oil Science and Technology, Academy of National Food and Strategic Reserves Administration, Beijing 102209, China; gmm@ags.ac.cn (M.G.); ylp3524@163.com (L.Y.); lixj@ags.ac.cn (X.L.); tanghuan0911@163.com (H.T.); lixinx11@163.com (X.L.); xyl@ags.ac.cn (Y.X.)

**Keywords:** rosemary extract, rapeseed oil, antioxidant effect, tocopherols, *tert*-butyl hydroxyquinone, storage stability test

## Abstract

Rapeseed oil is an important source of edible oil in the human diet and is also highly susceptible to oxidative deterioration. It has been demonstrated that rosemary extract (RE) can increase the oxidative stability of oils. In this work, the antioxidant capacity of rapeseed oil after the addition of RE during storage and the optimum addition of RE in rapeseed oil were investigated. Oxidative stability evaluation results demonstrate that the shelf life of rapeseed oil with the incorporation of 100 mg/kg of RE was equivalent to that with the addition of 50 mg/kg of *tert*-butyl hydroxyquinone (TBHQ). Storage test analysis results show that RE remarkably delayed the oxidation of rapeseed oil when the storage container was unsealed. The optimum amount of RE as an addition was 50–200 mg/kg under room temperature storage, while it was 150 mg/kg under Schaal oven storage. The antioxidant capacity of rapeseed oil with 50 mg/kg of RE added was remarkably higher than that with 50 mg/kg of TBHQ added after 20 d of storage, according to the Schaal oven test. Additionally, the addition of RE delayed the degradation of endogenous *α*-tocopherol in rapeseed oil. This study comprehensively evaluated the antioxidant properties of rapeseed oil when RE was added and it provides a new strategy for establishing healthy, nutritious, and safe oil preservation measures.

## 1. Introduction

Edible vegetable oils are rich in unsaturated fatty acids, which provide essential fatty acids for humans. However, oils with a high content of unsaturated fatty acids are highly susceptible to oxidative degradation during storage and processing [1], which not only reduces the content of essential fatty acids, but also produces some ingredients that are harmful to the human body [2,3]. Filling packaging with nitrogen or lipid antioxidants is the most commonly used preservation measure. Small-package edible vegetable oils (≤5 L) are often sealed with nitrogen, and oil oxidation happens when the container is unsealed due to exposure to air. In large-package oils, lipid antioxidants are often added, especially TBHQ, which is a synthetic antioxidant with a strong antioxidant property. Several studies have shown that synthetic antioxidants might have potential toxicity [4,5], and they are easy to evaporate or convert into harmful components at high temperatures [6], which may be related to their structures [7]. Therefore, at high temperatures, TBHQ has been found not only to reduce antioxidant performance, but also to be harmful to human health. In light of the above, more and more countries have begun to restrict or ban certain synthetic antioxidants [8]. To overcome these shortcomings, a growing number of studies are focusing on natural antioxidants [9].

Natural antioxidants are generally recognized as safe, and their applications in oils are less limited [10]. So, various studies have been performed to investigate the effect of natural antioxidants on the oxidative stability of vegetable oils [11,12]. For example, Vargas et al. [13] evaluated the efficacy of pitanga leaf extracts to protect canola oils against lipid oxidation, and found that the pitanga leaf extracts could preserve polyunsaturated fatty acids, preventing lipid oxidation during storage. Soybean oil supplemented with rambutan peel powder also showed positive effects on the retardation of the oxidation process during storage [14]. In addition to that, natural antioxidants are also increasingly being studied in the field of biodiesel [15,16]. Ahanchi et al. [17] found that 2500 mg/kg of pistachio hull extract was sufficient to increase the induction period of neat canola biodiesel from 1.53 h to above 3 h. In another study, walnut husk methanolic extract was added into waste cooking oil biodiesel, increasing the OSI of biodiesel by almost three times [18].

Rosemary (*Rosmarinus officinalis* L.) extract (RE) is also a natural antioxidant, and has antioxidant properties such as high efficiency and high temperature resistance. Its antioxidant effect is 2–4 times that of the synthetic antioxidants BHA and BHT. RE has a stable structure that is not easily decomposed at high temperatures and can withstand 190–240 °C [19,20]. The main advantage of RE is that it is more stable at high temperatures than most natural antioxidants [21]. In recent years, RE has been approved as a safe and effective antioxidant by many countries and regions, such as the European Union [22,23]. In China, RE was approved as a food additive antioxidant in 2016 (GB 1886.172-2016), and the maximum amount that can be added in vegetable oils is 700 mg/kg. More importantly, the high temperature resistance of RE makes it particularly suitable for edible oils used in thermal processing. Recently, studies on the antioxidant activity of RE in different types of oils have mainly focused on the oxidative stability of RE alone [24,25,26] or in combination with other synthetic or natural antioxidants [27]. It was indicated that RE could markedly delay the oxidation of oils. Significantly, the antioxidant effect of RE was molecularly superior to, or at least equivalent to, that of synthetic antioxidants commonly used in the food industry, such as BHA, BHT, and TBHQ [28]. When employed with other antioxidants, it could prominently improve the antioxidant efficacy of a single antioxidant, reduce the quantity of other antioxidants under given conditions, and so on [29,30]. Coincidentally, the Schaal oven and oxidative stability evaluation methods were often used as evaluation tools in these related studies. However, the room temperature storage method is rarely used in studies because of the longer storage period required.

Rapeseed oil is very popular in many regions, including Europe, China, the USA, and India. Following soybean oil and palm oil, rapeseed oil is the third most consumed vegetable oil around the world. According to the USDA, the world production of rapeseed oil reached 30.74 million tons in 2022/2023 (July), while the world consumption reached 30.08 million tons. Meanwhile, it also contains a high content of endogenous active components (such as tocopherols) [31]. However, reports about the effect of RE on the endogenous tocopherols in rapeseed oils and the relationships between RE content and the critical qualities of the oils are few. Hence, the influence of RE on the endogenous antioxidant tocopherols in rapeseed oil was determined using the Schaal oven test, and the relationship between RE content and oil quality was investigated. The results of this study provide a new strategy for establishing healthy, nutritious, and safe oil preservation measures.

## 2. Materials and Methods

### 2.1. Materials and Reagents

The rapeseed oil used was commercial refined oil obtained from a local supermarket, coupled with nitrogen packing. Rosemary extract (In the form of a brown powder, and was obtained via solvent extraction from rosemary stems and leaves. The content of CA is 85%) is one commercial product, and was purchased from a local supplier. *tert*-Butyl hydroxyquinone (TBHQ, ≥99%) was purchased from L&P Food Ingredient Co., Ltd. (Shaoguan, China). The analytical standards of TBHQ (≥98.0%), tocopherols, sterols, and fatty acid methyl esters were purchased from Sigma-Aldrich (Shanghai, China). HPLC-grade solvents including *n*-heptane, tetrahydrofuran, and methanol were purchased from Fisher (Shanghai, China). All other chemicals used in this study were analytical grade.

### 2.2. Samples Preparation

RE and TBHQ were added directly to rapeseed oil to obtain initial antioxidant-added oils (1000 mg/kg), including RE/rapeseed oil (RE/Oil) and TBHQ/rapeseed oil (TBHQ/Oil). The different concentrations of each antioxidant in rapeseed oils were prepared by diluting initial antioxidant-added oils with rapeseed oil, as shown in Table 1. Rapeseed oil without any additional antioxidants was used as the control sample.

### 2.3. Schaal Oven Test

Each sample in Table 1 was distributed into 7 glass beakers (7–8 cm × 9.5 cm i.d.), each containing about 120 g of oil, and placed in a thermostatically controlled oven at 65 ± 1 °C. The oil samples were shaken and then their positions were changed in the oven every day. The results of the pre-experiment indicate that the oil samples with different antioxidants added would have different degrees of oxidation susceptibility under the same storage, and the same sampling time would lead to insignificant oxidation processes of certain samples that are oxidized easily. Therefore, different sampling times were designed: the RE/Oil samples were sampled at 0, 5, 10, 15, 17, 19, and 20 d; the TBHQ/Oil samples were sampled at 0, 5, 10, 15, 20, 23, and 25 d; and the control samples were sampled at 0, 2, 4, 5, 6, 7, and 8 d. All the processed samples were stored at 4 °C for the scheduled analysis.

### 2.4. Room Temperature Storage Test after the Storage Container Is Unsealed

Each sample (4.5 kg) in Table 1 was packed in a 5-L PE bottle and stored at room temperature for 6 months. Under storage, the oil samples were exposed to air for 30 s and then shaken twice every day for the simulation of daily household use. After shaking, 120 g of the oil was collected every 6 d and stored at 4 °C for the scheduled analysis.

### 2.5. Quality Analysis of Rapeseed Oil

The determinations of fatty acid composition (GB 5009.168), tocopherol content (GB/T 26635/ISO 9936), sterol content (GB/T 25223/ISO 12228), AV (GB 5009.229), POV (GB 5009.227), and *P*-AV (GB/T 24304/ISO 6885) were carried out according to the standards.

### 2.6. Oxidative Stability Evaluation

The oxidative stability of oils was determined using a OSI-24 instrument (Plymouth, MN, USA) according to ISO 6886. Oxidative stability indices (OSI) were obtained in triplicate for all oil treatments, and corresponded to the break points in the plotted curves. The air flow rate was set at 10 L/h and the temperature of the heating block was maintained at 110 °C.

### 2.7. Determination of tert-Butyl Hydroxyquinone in Rapeseed Oil

The content of TBHQ in the oil was analyzed according to the procedure employed by Li et al. [32] with minor modifications. Briefly, 2.00 g of oil was accurately weighed and placed into a 50 mL centrifuge tube, then 4 mL of methanol was added to extract TBHQ. After extraction, the solutions were shaken vigorously for 2 min and centrifuged at 4000 r/min for 10 min. The supernatant was transferred into a 10 mL volumetric flask. The above procedures were repeated twice. The supernatant was combined until it reached exactly 10 mL and then was filtered, using a 0.45 μm organic filter membrane, for the HPLC analysis.

The HPLC equipment used in this study was a YiLiTe 3100 (Dalian, China) equipped with a UV detector. The stationary phase was a Sunfire C18 analytical column (250 × 4.6 mm i.d.) with a particle size of 5 μm (CA, USA) thermostated at 40 °C. The flow rate was 1.0 mL/min and the absorbance wavelength was 280 nm. The mobile phase was an isocratic elution consisting of acetic acid/water (1:99, 35%) and methanol (65%) for 25 min. The TBHQ was quantified by comparing the chromatographic areas of corresponding standards’ known concentrations.

### 2.8. Degradation of Tocopherols

The degradation rate of tocopherols was obtained by calculating the percent loss of each tocopherol isomer after heating the oil. It was calculated as follows:Degradation % = (*C_r_* − *C_h_*)/*C_r_* × 100%(1)
where *C_r_* is the content of each tocopherol isomer in raw rapeseed oil, and *C_h_* is the content of each tocopherol isomer in the heated oils.

### 2.9. Statistical Analysis

All results were expressed as mean ± standard deviation. The analysis was performed via one way analysis of variance (ANOVA) using Origin 2022 software (Northampton, MA, USA). Significance was declared for *p* < 0.05.

## 3. Results and Discussion

### 3.1. Quality Characteristics of Raw Rapeseed Oil

In order to clarify the antioxidant properties of rapeseed oils with RE added, some qualities and compositional parameters of raw rapeseed oil were analyzed. AV and POV are two safety limit indicators in vegetable oils; the rapeseed oils are classified into different quality grades according to these parameters. The AV and POV of raw rapeseed oil were 0.06 mg KOH/g oil and 0.33 mmol/kg, respectively, which met the first-level quality requirements of the China National Standard titled “Rapeseed oil (Grant No.: GB/T 1536-2021)”.

*P*-AV is usually used to evaluate the contents of secondary oxidation products in vegetable oils during processing and storage. In this study, the *P*-AV of raw rapeseed oil was 1.9.

Oleic acid was the major fatty acid (60.1 ± 0.36 g/100 g oil), followed by linoleic acid (19.0 ± 0.11 g/100 g oil), and linolenic acid (6.98 ± 0.01 g/100 g oil). These results demonstrate that the rapeseed oil had a high unsaturation degree. In this study, the rapeseed oil contained an abundance of active ingredients, such as tocopherols and phytosterols. The total content of tocopherols in raw rapeseed oil was 701.6 ± 17.9 mg/kg. *γ*-tocopherol was the main component (339.3 ± 1.0 mg/kg), followed by *α*-tocopherol (155.2 ± 0.4 mg/kg oil). Rapeseed oil also contains a variety of phytosterols. In this study, the total content of phytosterols in raw rapeseed oil was 571.79 ± 33.26 mg/100 g, with sitosterol and campesterol accounting for 55.89 ± 0.26% and 34.71 ± 0.10%, respectively.

In raw rapeseed oil, TBHQ and RE were undetectable in the HPLC analysis. It was suitable to study the antioxidant properties of exogenous antioxidants individually during storage.

### 3.2. Antioxidant Characteristics of Antioxidants in Rapeseed Oils

Worldwide food regulations have recognized RE as a natural and safe additive. In China, RE has been approved as a food additive, and the maximum amount that can be added in vegetable oil is 700 mg/kg according to GB 2760-2014, in which the limit of TBHQ added is 200 mg/kg. In order to compare and analyze the antioxidant capacity of rapeseed oils with RE or TBHQ added, the same addition gradient was designed within their respective maximum addition ranges (Table 1).

Preliminary assessments of the antioxidant capacity of rapeseed oils with different antioxidants added were performed using the oxidative stability evaluation method. This indicates that the concentration-dependent effects appeared (Figure 1): the OSI of RE/Oil and TBHQ/Oil increased with the addition amount. In addition, RE added in the rapeseed oil had a significant effect on prolonging the shelf life of oil, slightly weaker than the effect when TBHQ was added.

RE showed a prominent concentration-dependent effect on prolonging the shelf life from 50 to 700 mg/kg. When 50 mg/kg of RE was added in rapeseed oil, the OSI was 13.63 h, increasing by 3.77 h compared with the control sample. Meanwhile when 100 mg/kg of RE was added, the OSI was equivalent to that with the addition of 50 mg/kg of TBHQ. Similar to TBHQ, the OSI of rapeseed oil increased with the amount of RE added (Figure 1). When 700 mg/kg of RE was added, the OSI was 25.38 h, much higher than that of high-oleic canola oil with 1000 mg/kg of RE added [33]. This may be attributable to the differences between oil components and RE purity [34]. The CA content (85%) in RE in this work was much higher, which may also play an important role in prolonging the shelf life of rapeseed oils [35].

### 3.3. Effects of Antioxidants on the Quality of Rapeseed Oils in Schaal Oven Test

In order to investigate the antioxidant effect of RE on rapeseed oil more effectively, the Schaal oven test was used for evaluation, which can quickly evaluate the effect of antioxidants on oil quality during storage and obtain the optimum amount of RE added. According to the results of the pre-experiment in the Schaal oven test, TBHQ/Oil samples and RE/Oil samples had been stored for 25 and 20 d, respectively.

#### 3.3.1. Peroxide Value of Rapeseed Oils in Schaal Oven Test

POV was used as an indicator for the primary oxidation of rapeseed oils with different antioxidants during storage, and it is also one of the most widely used tests for measuring peroxides. A slower increase in POV implies a higher oxidative stability. The influence of antioxidants on the POV of rapeseed oil during storage is shown in Figure 2. Observably, it indicates that the POV of RE/Oil-1 stored for 10 d was 6.8 mmol/kg, much lower than that of the control sample (10 mmol/kg) stored for 4 d (Figure 2a). The increased amount of RE also had a similar effect; however, there was no significant difference (*p* < 0.05) in inhibiting the increase in POV among the samples with more RE added (RE/Oil-3—RE/Oil-7) during storage. It may be that the appropriate amount of RE in rapeseed oils was still effective enough to have antioxidant properties during storage.

The POV of all samples increased with the storage time, and increased with the acceleration of the samples with a low antioxidant amount (TBHQ/Oil-1 and RE/Oil-1) after 10 d of storage, also appearing in the TBHQ/Oil-2 and RE/Oil-2 samples during later storage. However, the POVs of the other samples increased slowly during storage and were not significantly different (*p* < 0.05) among samples with different addition amounts of the same additive (RE/Oil-3—RE/Oil-7 and TBHQ/Oil-3—TBHQ/Oil-4). Therefore, the optimum amount of RE as an addition in rapeseed oil was about 150 mg/kg within 20 d of storage, according to the Schaal oven test. TBHQ remained the most effective and gave the lowest POV under the same storage conditions (Figure 2b), but its effect was weaker than that of RE added during later storage with a low amount added (50 mg/kg).

#### 3.3.2. P-Anisidine Value of Rapeseed Oils in Schaal Oven Test

*P*-AV is the measure of the secondary oxidation products (such as aliphatic aldehydes, ketones, alcohols, acids, and hydrocarbons) generated during the decomposition of hydroperoxides. It is based on the reaction of the aldehyde carboxy bond on the *P*-anisidne amine group, leading to the formation of a Schiff base that absorbs at 350 nm [36]. A lower *P*-AV indicates that a less rancid oil has been produced. The results of *P*-AV were similar to the POVs of samples. The *P*-AV of all samples increased with storage time (Figure 3), especially during the later storage, and the *P*-Avs of the control sample, RE/Oil-1, TBHQ/Oil-1, and TBHQ/Oil-2 increased with acceleration. The results also demonstrate that RE and TBHQ could cause a significant reduction in the *P*-AV of rapeseed oils, and that the effect of TBHQ was better than that of RE under the same storage. But, in a low amount (50 mg/kg), the effect of RE was better than that of TBHQ after 15 d of storage. In the continuous heating process, TBHQ is degraded or transformed [7], resulting in a reduction in the components that played an antioxidant role in the oil. Therefore, during the later heating storage, when the amount of antioxidants in rapeseed oils was low, the oxidation products of the oils were increased significantly, resulting in a significant increase in oxidation indicators (POV and *P*-AV).

#### 3.3.3. Content Changes of Tocopherols in Heated Rapeseed Oils during Storage

Rapeseed oils contain the endogenous antioxidants tocopherols, which can slow down oil oxidation. Rapeseed oils are also a natural and convenient source of vitamin E for humans. The data from this study show that the exogenous antioxidant RE not only protected oil quality, but also played an important role in protecting endogenous tocopherols, which is beneficial to human health. During 65 °C storage, the loss of tocopherol isoforms in heated rapeseed oil was found to be isoform-dependent, with *α*-tocopherol consistently being degraded to the greatest extent, followed by *γ*-tocopherol. This mainly indicates that the degradation of *α*-tocopherol was higher than that of *γ*-tocopherol, which the results showed: *α*-tocopherol in the control sample stored for 7 d was almost completely degraded, whereas *γ*-tocopherol degradation was about 50% (Figure 4a). Elisia et al. [37] investigated the degradation of tocopherol isoforms in seven kinds of vegetable oils heat-treated at 95 °C for 1 d and concluded that *α*-tocopherol degradation was faster than that of *γ*-tocopherol. When canola oil was employed as an example, the degradation of *α*-tocopherol was 52.4%, while for *γ*-tocopherol it was 12.2%. It is evidenced that tocopherols, especially *α*-tocopherol, are susceptible to the heating process.

However, the presence of RE in rapeseed oil significantly (*p <* 0.05) reduced the loss of tocopherols during storage at 65 °C. The degradations of *α*-tocopherol and *γ*-tocopherol in RE/Oil-1 stored for 10 d were almost similar to that in the control sample stored for 2 d. When stored for 17 d, the degradation rate of *α*-tocopherol in RE/Oil-1 reached 100%, which was effectively delayed by 10 d compared with the control sample. Similarly, Kitts et al. [38] reported that 0.1% RE added could reduce the loss of *γ*-tocopherol in both soybean and hempseed oil during the heating process. This may be attributed to the protection effect of RE on tocopherols during the heating process or storage, which requires further studies.

The degradation of tocopherols decreased with an increased amount of RE (Figure 4b). When the amount of RE as an addition was 150 mg/kg (RE/Oil-3) and the oils were stored for 20 d, the degradation rates of *α*-tocopherol and *γ*-tocopherol decreased to 9.9 ± 0.5% and 5.9 ± 0.8%, respectively, which were significantly lower than the rates with 50 mg/kg of RE added (RE/Oil-1) and stored for 15 d. The results imply that RE had a remarkable effect on the degradation of tocopherols, and the increased amount of RE could prevent the large losses of tocopherols. However, the degradation of tocopherols was no longer significantly altered by the further addition of RE (RE/Oil-5 and RE/Oil-7). This may be related to the storage time, so that the excessive amount of RE in rapeseed oil could not play a protective effect under short storage. Additionally, the maximum amount of RE may be not the optimum amount to inhibit tocopherol degradation during storage, and a suitable amount of RE was 150 mg/kg, consistent with the result of the POV. Meanwhile, the study found that, when stored for 15 d, the samples with low addition (RE/Oil-1 and TBHQ/Oil-1) had a similar POV, but the RE/Oil-1 protected endogenous tocopherols slightly better from degradation than TBHQ/Oil-1 (Figure 4a). However, the reports are few, and to know whether RE played a role in regenerating oxidized tocopherol molecules [39] will require further studies.

Similar to RE, TBHQ also had a significant effect on decreasing the degradation of tocopherols. When stored for 20 d, the degradation rates of *α*-tocopherol and *γ*-tocopherol in TBHQ/Oil-1 (50 mg/kg of TBHQ) reached 100%, while those of *α*-tocopherol and *γ*-tocopherol in TBHQ/Oil-2 (100 mg/kg of TBHQ) decreased to 6.2 ± 2% and 2.6 ± 1.4%. Meanwhile, the POV of TBHQ/Oil-1 also showed a significant upward trend during storage, far higher than the limit value (10 mmol/kg) after 20 d of storage, but TBHQ/Oil-2 was lower than the limit value. This phenomenon indicates that TBHQ was similar to RE, and had weakening antioxidant properties during the long-term heating process, resulting in the destruction of the endogenous tocopherols and oil quality deterioration. However, increasing the amount of RE and TBHQ could protect endogenous tocopherols and improve oil quality.

### 3.4. Effects of Antioxidants on the Quality of Rapeseed Oil Stored at Room Temperature

The room temperature storage test can reflect the quality characteristics of unsealed vegetable oils before daily consumption. In this study, we investigated the effect of antioxidants on the quality of rapeseed oil stored for 6 months at room temperature.

#### 3.4.1. Peroxide Value of Rapeseed Oils Stored at Room Temperature

The influence of antioxidants on the POV of rapeseed oil samples under 6 months of storage is shown in Figure 5. The initial POV was 0.33 ± 0.02 mmol/kg, meeting the first-level quality requirement (≤5 mmol/kg) of GB1536-2021 (Chinese Standard titled “Rapeseed oil”). After 5 months of storage, the POV of rapeseed oil after the addition of RE was 1.6–2.0 mmol/kg, and that of the control sample was 1.8 mmol/kg. However, the POVs were 2.2–3.0 mmol/kg and 9.2 mmol/kg after 6 months of storage, respectively (Figure 5a). The POV of the control sample stored for 6 months had already reached the second- and third-level quality requirements (≤10 mmol/kg) of GB1536-2021, similar to the result obtained by Liu et al. [40], who reported that the refined rapeseed oil could be stored for about 196 d at 25 °C with a POV of 10.38 mmol/kg. In this study, rapeseed oil with RE added showed obvious antioxidant characteristics, but the antioxidant effect was weaker than that with TBHQ added. Martinez et al. [30] also found that the inhibitory effect of sunflower oil with 100 mg/kg of TBHQ added on the increase in POV was significantly better than that with 800 mg/kg of RE added after 6 months of storage, according to the room temperature test. Unexpectedly, unlike the TBHQ, the POV of RE/Oil samples did not show a decreasing trend with an increasing amount of RE, nor an equivalent decrease caused by doubling the addition amount. This result suggests that the optimum amount of RE in rapeseed oil could be 50–200 mg/kg, similar to that obtained in the Schaal oven test. The result from Redondo et al. [41] showed that the optimum amount of rosemary powder could be 1–1.5%, and that the difference may be due to differences in the composition of the antioxidant and the CA content. Therefore, different Res may lead to different antioxidant effects.

All the POV data were used to perform polynomial regression equations with storage time as the independent variable (Table 2). This can predict the storage time under room temperature. When the POV of samples reached 1.8 mmol/kg (the POV of the control sample in 5 months was 1.8 mmol/kg), the TBHQ/Oil samples could be stored for 182.9–210.7 d, while the RE/Oil samples (RE/Oil-1-RE/Oil-4) could be stored for 157.4–161.0 d, shorter than the TBHQ/Oil samples. However, the maximum amount of antioxidants did not seem to be the best option. For RE/Oil samples, the storage time after which the POV reached 1.8 mmol/kg was shortened with an increased amount of RE, and a low addition of RE was more beneficial for the storage of unsealed rapeseed oils at room temperature. However, the related reports are few, and more studies will be needed to validate this finding.

#### 3.4.2. P-Anisidine Value of Rapeseed Oils Stored at Room Temperature

In this study, the unsealed oil samples were stored for 6 months at room temperature, and the *P*-AV results showed no obvious changes with the prolongation of storage time. TBHQ and RE had no apparent effect on inhibiting the formation of secondary oxidation products. This result might be due to the short storage time and the stable structure of rapeseed oil. The quality of rapeseed oil had been less affected under the storage environment, and there was a lack of secondary oxidation product buildup during the 6 months of storage.

## 4. Conclusions and Prospects

RE is a natural, high-temperature-resistant antioxidant and it is used as a food additive. Herein, the application of RE in the rapeseed oil used as a high temperature edible oil was investigated. The results show that the unsealed rapeseed oil with the addition of RE has the apparent effect of prolonging the consumption time of oil, based on a comprehensive analysis of two tools for evaluating the oxidative stability of oils. Much more importantly, the optimum amount of RE as an addition was 50–200 mg/kg under room temperature, while it was 150 mg/kg under Schaal oven storage. In light of the above, these results may provide a reasonable guide on the amount of RE that should be added in daily-use oils, and on a safe consumption time for consumers. This may be more targeted and applicable than other studies on the antioxidant properties of RE added in oils. In addition, the results also show that the antioxidant capacity of rapeseed oil with RE (50 mg/kg) added was significantly higher than that with TBHQ added after 20 d of storage, according to the Schaal oven test. Therefore, RE can be used as an alternative to TBHQ, BHA, and BHT, i.e., in rapeseed oil under certain conditions. We also found that the exogenous antioxidant RE in rapeseed oil not only protected the quality of the oil, but also played an important role in protecting the nutrient tocopherols in rapeseed oil. The rapeseed oil with 50 mg/kg of RE added could delay the complete degradation of endogenous *α*-tocopherol by 10 d. This new strategy for establishing healthy, nutritious, and safe oil preservation measures may provide a rich and effective nutrient source in the human diet, and will be beneficial to human health.

## Figures and Tables

**Figure 1 foods-12-03583-f001:**
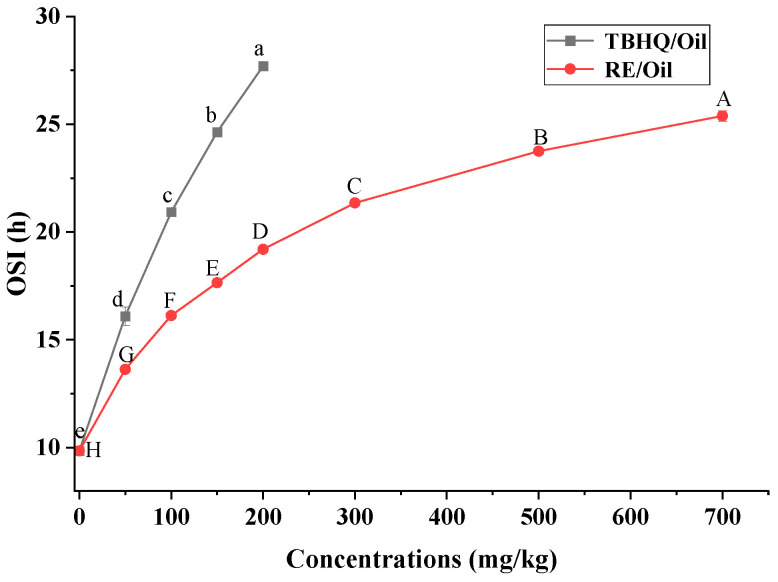
Effects of antioxidant concentrations on OSI. Plotted values were means of two independent determinations. Bars represent standard deviation. Different letters in one curve are significantly different (*p* < 0.05); meanwhile, the same letters are not significantly different (*p* < 0.05). Regression equation of RE/Oil samples *y* = 11.04 + 0.04678 *x* − 3.860 × 10^−5^ *x*^2^, where ‘*y*’ is the dependent variable (protection factor); ‘*x*’ is the independent variable (RE concentration). Determination coefficient: *R*^2^ = 0.98.

**Figure 2 foods-12-03583-f002:**
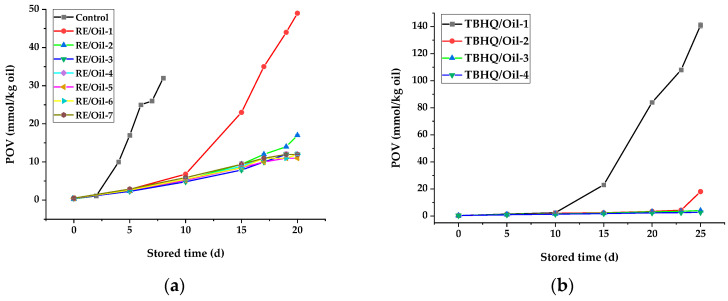
Effects of antioxidants on POV of rapeseed oil stored at 65 °C. POV of oils in presence of (**a**) control and RE/Oil samples and (**b**) TBHQ/Oil samples shown in Table 1.

**Figure 3 foods-12-03583-f003:**
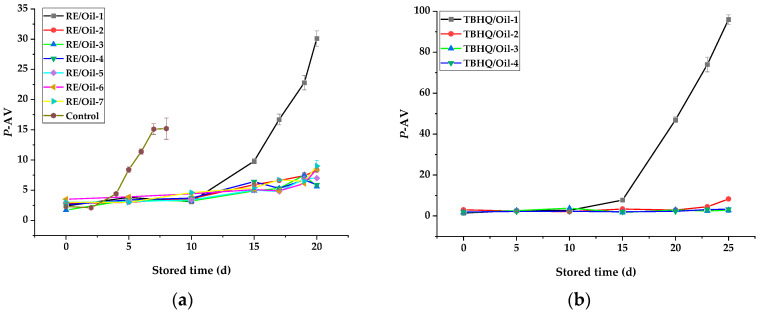
Effects of antioxidants on *P*-AV of rapeseed oils stored at 65 °C. *P*-AV of oils in presence of (**a**) control and RE/Oil samples and (**b**) TBHQ/Oil samples shown in Table 1.

**Figure 4 foods-12-03583-f004:**
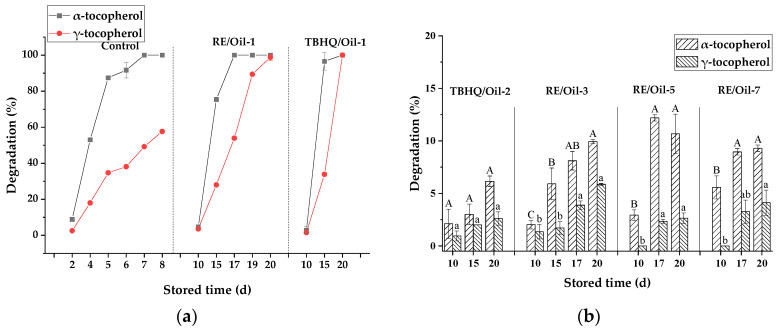
Content changes of tocopherols in heated rapeseed oils. Tocopherol degradation of oils in the presence of (**a**) control, RE/Oil-1 and TBHQ/Oil-1, (**b**) TBHQ/Oil-2, RE/Oil-3, RE/Oil-5, RE/Oil-7, shown in Table 1. In (**b**), different letters in one sample are significantly different (*p* < 0.05); meanwhile, the same letters are not significantly different (*p* < 0.05).

**Figure 5 foods-12-03583-f005:**
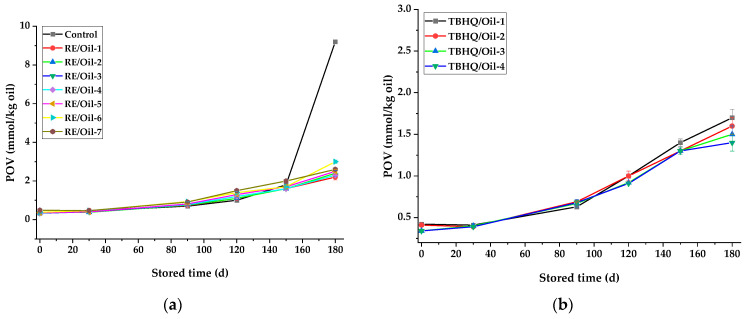
Effects of antioxidants on the POV of unsealed samples stored at room temperature. POV of oils in the presence of (**a**) control and RE/Oil samples and (**b**) TBHQ/Oil samples shown in Table 1.

**Table 1 foods-12-03583-t001:** Concentrations of antioxidants in rapeseed oils.

Oil Samples	Concentrations (mg/kg)							
	0	1	2	3	4	5	6	7
RE/Oil	0	50	100	150	200	300	500	700
TBHQ/Oil	0	50	100	150	200	/	/	/

**Table 2 foods-12-03583-t002:** Regression coefficients and determination coefficients (*R*^2^) for POV from unsealed rapeseed oils during the storage stability test. See Table 1 for oil samples.

Oil Samples	Regression Coefficients			*R* ^2^	Estimated Time (d) for POV = 1.8 mmol/kg
	*β* _0_	*β* _1_	*β* _2_		
Control	0.3982	−0.00515	0.00009342	0.9447	153.1
TBHQ/Oil-1	0.4029	−0.0008988	0.00004668	0.9816	182.9
TBHQ/Oil-2	0.3796	0.000631	0.00003516	0.9900	192.2
TBHQ/Oil-3	0.3301	0.00157	0.00002878	0.9836	200.4
TBHQ/Oil-4	0.3164	0.00255	0.00002132	0.9636	210.7
RE/Oil-1	0.3886	−0.00171	0.00006508	0.9928	161.0
RE/Oil-2	0.3857	−0.00362	0.00008095	0.9952	156.4
RE/Oil-3	0.3540	−0.00113	0.00006553	0.9963	157.4
RE/Oil-4	0.3723	−0.00118	0.000065106	0.9956	157.4
RE/Oil-5	0.3691	−0.00155	0.00007328	0.9938	150.7
RE/Oil-6	0.4394	−0.00420	0.000097433	0.9832	153.9
RE/Oil-7	0.4558	−0.000369	0.00006955	0.9915	141.7

Regression equations *y = β*_0_
*+ β*_1_*x + β*_2_*x*^2^, where ‘*y*’ is the dependent variable (peroxide value/POV); *β*_0_ is a constant that it is equal the value of ‘*y*’ when the value of ‘*x*’ = 0; *β*_1_ is the coefficient of ‘*x*’; *β*_2_ is the coefficient of ‘*x*^2^*’*; ‘*x*’ is the independent variable (time).

## Data Availability

The data that support the findings of this study are available on request from the corresponding author.

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
