# Peer review of "Antioxidant Efficacy of Rosemary Extract in Improving the Oxidative Stability of Rapeseed Oil during Storage"

_foods, 2023, doi:10.3390/foods12193583_

Round 1

Reviewer 1 Report

This paper needs major revisions.

1)      The quality of both language and organization needs to be improved by proofreading by a native English speaker or proofreading service. For example, the title should be “Antioxidant efficacy of rosemary extract in improving the oxidative stability of rapeseed oil during storage”. 2)      Please extend the abstract a bit by including more numerical results. 3)      Do not use abbreviations in keywords. Add one more keyword.

4)      Please add a Table of Abbreviations/Nomenclatures/Symbols.

5)      Please avoid reference lumping.

6)      The introduction section is short and does not include state-of-the-art of research in this field.

7)      The novelty/originality of the paper should be more effectively established. It would be advisable to add a Table to the “Introduction” section, tabulating the latest research works in the field to highlight the novelty of the present work accordingly.

8)      The latest research trends and reviews on using natural antioxidants in improving the oxidative stability of vegetable oils and their derivatives (e.g., biodiesel) should be included and discussed to enrich the Introduction. Here are some examples, which, if found useful by the authors, can be used: “Biodiesel antioxidants and their impact on the behavior of diesel engines: A comprehensive review”, “Unlocking the potential of walnut husk extract in the production of waste cooking oil-based biodiesel”, and “Pistachio (Pistachia vera) wastes valorization: enhancement of biodiesel oxidation stability using hull extracts of different varieties”. Authors can briefly discuss this issue using works such as the example provided, but not necessarily limited to that, and highlight its importance.

9)      Please avoid having heading after heading with nothing in between; either merge your headings or provide a small paragraph in between.

10)  Uncertainty analysis is required for experimental works.

11)  Please check all units. For example, “days” should be “d”. Make sure all the units will be presented in compliance with the SI System.

12)  Please reduce the significant figures of the data reported in the paper and figures (Maximum four). Here is an example of significant figures (sig figs):

a.       10082 (5 sig figs)

b.      70,000 (1 sig fig)

c.       0.0025 (2 sig figs)

d.      0.000309 (3 sig figs)

e.       50010.000 (8 sig figs)

13)  Some paragraphs are too short or too long. This negatively impacts the structure of the manuscript.

14)  Avoid using abbreviations and symbols in the headings.

15)  The section “Results and Discussion” is too poor. Deep and mechanistic discussions are required to explain the result reported in the literature.

16)  Discussions have not been well supported by proper references.

17)  Add practical implications of the study.

18)  Limitations of the study should be included and discussed.

19)  The obtained results have not been sufficiently compared with the published data. Please add a Table in the “Results and Discussion” section to address this issue.

20)  Future studies should investigate the sustainability features of the production of bioproducts (e.g., natural antioxidants) using advanced sustainability assessment tools, including life cycle assessment and exergy, as elaborated in recent work such as “The role of sustainability assessment tools in realizing bioenergy and bioproduct systems”, “Environmental life cycle assessment of different biorefinery platforms valorizing olive wastes to biofuel, phosphate salts, natural antioxidant, and an oxygenated fuel additive (triacetin)”, etc. Please briefly discuss this future research need using works such as the examples provided, but not necessarily limited to them, and highlight the importance of such additional assessments to direct future studies.

21)  Please change “Conclusions” to “Conclusions and Prospects”. This part simply presents the results obtained throughout the study.

This paper needs major revisions before publication in “Foods”.   1)      The quality of both language and organization needs to be improved by proofreading by a native English speaker or proofreading service. For example, the title should be “Antioxidant efficacy of rosemary extract in improving the oxidative stability of rapeseed oil during storage”. 2)      Please extend the abstract a bit by including more numerical results. 3)      Do not use abbreviations in keywords. Add one more keyword.

4)      Please add a Table of Abbreviations/Nomenclatures/Symbols.

5)      Please avoid reference lumping.

6)      The introduction section is short and does not include state-of-the-art of research in this field.

7)      The novelty/originality of the paper should be more effectively established. It would be advisable to add a Table to the “Introduction” section, tabulating the latest research works in the field to highlight the novelty of the present work accordingly.

8)      The latest research trends and reviews on using natural antioxidants in improving the oxidative stability of vegetable oils and their derivatives (e.g., biodiesel) should be included and discussed to enrich the Introduction. Here are some examples, which, if found useful by the authors, can be used: “Biodiesel antioxidants and their impact on the behavior of diesel engines: A comprehensive review”, “Unlocking the potential of walnut husk extract in the production of waste cooking oil-based biodiesel”, and “Pistachio (Pistachia vera) wastes valorization: enhancement of biodiesel oxidation stability using hull extracts of different varieties”. Authors can briefly discuss this issue using works such as the example provided, but not necessarily limited to that, and highlight its importance.

9)      Please avoid having heading after heading with nothing in between; either merge your headings or provide a small paragraph in between.

10)  Uncertainty analysis is required for experimental works.

11)  Please check all units. For example, “days” should be “d”. Make sure all the units will be presented in compliance with the SI System.

12)  Please reduce the significant figures of the data reported in the paper and figures (Maximum four). Here is an example of significant figures (sig figs):

a.       10082 (5 sig figs)

b.      70,000 (1 sig fig)

c.       0.0025 (2 sig figs)

d.      0.000309 (3 sig figs)

e.       50010.000 (8 sig figs)

13)  Some paragraphs are too short or too long. This negatively impacts the structure of the manuscript.

14)  Avoid using abbreviations and symbols in the headings.

15)  The section “Results and Discussion” is too poor. Deep and mechanistic discussions are required to explain the result reported in the literature.

16)  Discussions have not been well supported by proper references.

17)  Add practical implications of the study.

18)  Limitations of the study should be included and discussed.

19)  The obtained results have not been sufficiently compared with the published data. Please add a Table in the “Results and Discussion” section to address this issue.

20)  Future studies should investigate the sustainability features of the production of bioproducts (e.g., natural antioxidants) using advanced sustainability assessment tools, including life cycle assessment and exergy, as elaborated in recent work such as “The role of sustainability assessment tools in realizing bioenergy and bioproduct systems”, “Environmental life cycle assessment of different biorefinery platforms valorizing olive wastes to biofuel, phosphate salts, natural antioxidant, and an oxygenated fuel additive (triacetin)”, etc. Please briefly discuss this future research need using works such as the examples provided, but not necessarily limited to them, and highlight the importance of such additional assessments to direct future studies.

21)  Please change “Conclusions” to “Conclusions and Prospects”. This part simply presents the results obtained throughout the study.

Reviewer 2 Report

Dear Colleagues

I have some observations regarding

1. the rosemary extract used in the study was obtained by you, by what method and if it was characterized

2. What method was used to determine the degradation of tocopherols

3. How did the profile of fatty acids in rapeseed oil change during storage?

Author Response

Please see the attachmen.

Reviewer 3 Report

This manuscript is on the application of rosemary extract (RE) as an natural antioxidant in the rapeseed oil in comparison with TBHQ.

The abstract should be revised as it is written in a unscientific way. In the first line it is written that RE has antioxidative properties, ok! then what was the reason to do this research. The abstract should be started with the rapeseed oil low oxidative stability and the need to use antioxidants. Line 10, "... from different addition of.." different of what? concentration? amount or level?. Please revise the whole abstract on the English writing. Abstract should be informative and presenting some data not just expressing the qualitative results.

In the method section, the sampling for oil containing RE and TBHQ and control samples were done in different times, it can affect the statistical works and anlaysis.

Line 183, "on delaying the shelf life", what does it mean? May be it was enhancing and prolonging the shelf life?

OSI of oil containing TBHQ always was higher than the OSI of oil contaning rapeseed oil, but it was concluded that RE can be alternative to TBHQ or even better. Please see the Figure 1.

Fig 2 and Fig 4 there is no statistical works on the obtained data and results. It could be better to use Tables instead of figure.

Very complex sentences in the discussion part, for example: line 282-284:The results implied that RE had a remarkable effect on the degradation rate of tocopherol, however, the decrease of degradation rate was no longer increased with the amount of RE continuing to increase (RE/Oil-5 and RE/Oil-7). What does this statement mean? Please check the whole discussion part and revise the writing style.

There is no comparing the obtained results withe the previously published data.

Writing in some cases are complex and very poor and needs very serious revision to be smooth and easy to follow. Some examples were written in the comments to author section.

Round 2

Reviewer 1 Report

This paper has been throughly revised according to the comments given by the reviewers and can be published in its present form.

Author Response

Thank you very much for taking the time to review this manuscript. We have revised the manuscript as academic editor required.Please see the attachment.

Reviewer 3 Report

Manuscript was revised according to the comments and its acceptance is recommended.

Author Response

(The authors gave the same response as above.)
